# Second Generation Small Molecule Inhibitors of Gankyrin for the Treatment of Pediatric Liver Cancer

**DOI:** 10.3390/cancers14133068

**Published:** 2022-06-22

**Authors:** Amber M. D’Souza, Manu Gnanamony, Maria Thomas, Peter Hanley, Dipti Kanabar, Pedro de Alarcon, Aaron Muth, Nikolai Timchenko

**Affiliations:** 1Department of Pediatrics, University of Illinois College of Medicine Peoria, 1 Illini Drive, Peoria, IL 61605, USA; manug@uic.edu (M.G.); tmaria@uic.edu (M.T.); pjhanle2@uic.edu (P.H.); pdealarc@uic.edu (P.d.A.); 2Department of Pharmaceutical Sciences, St. John’s University, 8000 Utopia Pkwy, Jamaica, NY 11439, USA; dipti.kanabar17@my.stjohns.edu (D.K.); mutha@stjohns.edu (A.M.); 3Division of Pediatric Surgery, Cincinnati Children’s Hospital Medical Center, 3333 Burnet Avenue, Cincinnati, OH 45229, USA; nikolai.timchenko@cchmc.org

**Keywords:** Gankyrin, cjoc42 derivative, hepatoblastoma, pediatric hepatocellular carcinoma, therapy, drug synergy

## Abstract

**Simple Summary:**

New therapy options are needed for children with liver cancer. The goal of this study was to evaluate the role of three new compounds in the treatment of liver cancer cells. These compounds all inhibit a protein called Gankyrin, which is known to promote liver cancer by destroying tumor suppressor proteins. We demonstrated that liver cancer cells have significantly reduced proliferation when treated with these compounds by preventing the degradation of tumor suppressor proteins. We also discovered that these compounds enhance the effects of doxorubicin, which is a chemotherapy drug commonly used in liver cancer. These results support continued efforts in Gankyrin-based therapy for the treatment of pediatric liver cancer.

**Abstract:**

Background: Gankyrin, a member of the 26S proteasome, is an overexpressed oncoprotein in hepatoblastoma (HBL) and hepatocellular carcinoma (HCC). Cjoc42 was the first small molecule inhibitor of Gankyrin developed; however, the IC_50_ values of >50 μM made them unattractive for clinical use. Second-generation inhibitors demonstrate a stronger affinity toward Gankyrin and increased cytotoxicity. The aim of this study was to characterize the in vitro effects of three cjoc42 derivatives. Methods: Experiments were performed on the HepG2 (HBL) and Hep3B (pediatric HCC) cell lines. We evaluated the expression of TSPs, cell cycle markers, and stem cell markers by Western blotting and/or real-time quantitative reverse transcription PCR. We also performed apoptotic, synergy, and methylation assays. Results: The treatment with cjoc42 derivatives led to an increase in TSPs and a dose-dependent decrease in the stem cell phenotype in both cell lines. An increase in apoptosis was only seen with AFM-1 and -2 in Hep3B cells. Drug synergy was seen with doxorubicin, and antagonism was seen with cisplatin. In the presence of cjoc42 derivatives, the 20S subunit of the 26S proteasome was more available to transport doxorubicin to the nucleus, leading to synergy. Conclusion: Small-molecule inhibitors for Gankyrin are a promising therapeutic strategy, especially in combination with doxorubicin.

## 1. Introduction

Pediatric liver cancer represents 2% of childhood malignancies, with the two most common types being hepatoblastoma (HBL) and hepatocellular carcinoma (HCC) [1]. Chemotherapy remains a cornerstone of therapy; however, chemoresistance remains an issue [2,3]. About 30% of patients with HBL will eventually relapse or die of chemoresistant disease, and only about 40% of children with HCC will show any response to chemotherapy [3,4,5,6]. A novel therapy is desperately needed to improve the outcomes in children with liver cancer.

The use of targeted, small-molecule therapy has been gaining traction in the oncology world [7]. In theory, drugs that target specific molecular or genetic alterations in tumors have a more potent anticancer effect while minimizing unwanted side effects. One of the key events in the development of HBL is the elimination of proteins that protect the liver from cancer development [8,9,10]. Prior work in liver tumor biology has demonstrated Gankyrin as a driver oncoprotein that is elevated in HBL and HCC [11,12,13]. This oncoprotein is part of the 26S proteasome that promotes tumor growth by delivering tumor suppressor proteins (TSPs), including p53, Retinoblastoma (Rb), CUGBP1, C/EBPα, and HNF4α, to the proteasome for degradation [12]. Additionally, Gankyrin has been found to be overexpressed in a variety of other solid tumors and linked to chemoresistance [14,15,16]. These biological studies suggest that Gankyrin is a promising drug target for pediatric liver cancer.

Cjoc42 was the first small-molecule inhibitors of Gankyrin generated [17]. This drug was shown to inhibit the function of Gankyrin, likely through a conformational change in the protein that impairs its ability to bind and degrade TSPs [8]. In vitro studies demonstrated an antiproliferative effect in pediatric liver cancer; however, the IC50 values were >50 μM, making it unattractive for clinical practice [8].

More recently, second-generation Gankyrin inhibitors based on modifications to the cjoc42 structure have been developed (Figure 1) [18,19]. These inhibitors have a stronger affinity toward Gankyrin based on binding assays and better IC50 values in pediatric liver cancer cell lines (Figure 2). The purpose of our study is to characterize the in vitro effects of three cjoc42 derivatives: AFM-1-2, DK-1–7, and JA-1–38. Additionally, given that the targeted drugs tend to work best with traditional chemotherapy rather than as single agents, we determined if there is drug synergy between the cjoc42 derivatives and cisplatin or doxorubicin, two commonly used chemotherapeutics in liver cancer.

## 2. Materials and Methods

### 2.1. Cjoc42 Derivatives and Chemotherapy Agents

Cjoc42 derivatives were supplied by Dr. Aaron Muth’s lab and have been previously described [18]. These derivatives were initially dissolved in 100% DMSO to a concentration of 10 mM. For the cell culture experiments, they were further diluted in cell culture media to a concentration of 0.1–1% DMSO.

Cisplatin (Medchem Express; Monmouth Junction, NJ, USA) was prepared as a 3.3-mM stock in sterile water, and doxorubicin (Medchem Express) was dissolved in dimethyl sulfoxide (DMSO) to a concentration of 100 μM. Both drugs were stored at −80 °C in single-use aliquots to prevent freeze–thaw cycles. For experiments, the drugs were diluted in cell culture media to achieve the needed concentration. 

### 2.2. Cell Culture Studies

HepG2 (HB-8065) and Hep3B (HB-8064) cells were purchased from the ATCC. After being obtained, they were immediately amplified, with many early passages frozen for subsequent studies. HepG2 and Hep3B cells were grown in Dulbecco’s Modified Eagle’s Medium (DMEM, Fisher; Waltham, MA, USA). All media was supplemented with 10% fetal bovine serum (FBS, VWR; Vienna, Austria), 1X non-essential amino acids (Corning; Corning, NY, USA), and 1% penicillin/streptomycin (Corning; Corning, NY, USA). Cells were incubated at 37 °C in a CO_2_ incubator. Mycoplasma contamination was monitored using a Mycoplasma PCR Detection Kit & Elimination (Abm; Richmond, BC, Canada).

### 2.3. Proliferation Assays

HepG2 and Hep3B cells were seeded in triplicate in 96-well plates at 5.0 × 10^4^ cells per well. After 24 h, cells were treated with various concentrations of each cjoc42 derivative (5–25 μM). After 72 h of treatment, an MTT assay was performed. Briefly, the treatment media was removed, and the cells were incubated at 37 °C with 100 µL of 0.5 mg/mL MTT reagent dissolved in DMEM. After 2 h, the MTT reagent was removed, followed by the addition of 100 µL of DMSO. The cells were incubated again for 10 min at 37 °C, and the plates were read at 570 nm using a spectrophotometer.

### 2.4. Saturation Assay

HepG2 and Hep3B cells were seeded in triplicate in 96-well plates at 2.0 × 10^4^ cells and 3.0 × 10^4^ per well, respectively. After 24 h, the cells were treated with various concentrations of each cjoc42 derivative (0.2, 1, 5, 25, 125, and 625 μM). The MTT assay was performed at 24 h post the drug addition. Briefly, the treatment media was removed, and the cells were incubated at 37 °C with 100 µL of 0.5 mg/mL MTT reagent dissolved in DMEM. After 2 h, the MTT reagent was removed, followed by the addition of 100 µL of DMSO. The cells were incubated again for 10 min at 37 °C, and the plates were read at 570 nm using a spectrophotometer.

### 2.5. Protein Isolation and Western Blotting

Nuclear and cytoplasmic extracts were prepared as previously shown [20,21]. The extracts (5–17 μg) were fractionated using SDS PAGE and transferred to nitrocellulose membranes (Bio-Rad). Membranes were probed with the corresponding antibodies. The following antibodies were used: CUGBP1 (sc-20003), HNF4α (sc-374229), Rb (sc-102), p53 (sc-126), cdc2 (sc-54), phosphorylated Rb (9307S), 20S proteasome α1/α2/α3/α4/α5/α6/α7 (sc-58412), β-actin (sc-47778, 20536-1-AP), and HDAC1 (34589S). The membranes were then incubated with the corresponding secondary antibodies (sc-516102, 7074S, and 7076S). The results were imaged on X-ray films and quantified using ImageJ 1.52a software (National Institute of Health, Bethesda, MD, USA). The experiment was performed in triplicate, and the mean ± SEM values were used for statistical analysis. All original western blot images can be found in the Appendix A.

### 2.6. RNA Extraction and Real-Time Quantitative Reverse Transcriptase PCR

Total RNA was extracted using the Trizol reagent (Invitrogen, Waltham, MA, USA). The RNA concentration was quantified using Nanodrop 2000 (Thermo Scientific; Waltham, MA, USA). Complementary DNA (cDNA) was synthesized with 1 μg of total RNA using a High-Capacity cDNA Reverse Transcription kit (Applied Biosystems, Waltham, MA, USA) according to the manufacturer’s instructions. The cDNA was diluted 5 times with nuclease-free water. 

Relative quantification of the mRNA was performed by real-time PCR using the PowerUp SYBR Green Master Mix (Applied Biosystems). A 10-μL reaction mix containing 1 μL each of the forward and reverse primers, 5 μL of 2× PowerUp SYBR Master Mix, 1 μL nuclease-free water, and 2 μL of the diluted cDNA template was used for the amplification of each target gene. The cycling conditions used were as recommended by the kit manufacturer. All the primers used for this study were purchased from Integrated DNA Technologies, USA. The primer sequences are given in Appendix A. HPRT was used as the internal control (housekeeping gene), and the fold changes were calculated using the 2^−ΔΔCT^ method [22].

### 2.7. Wound Healing Assay

Hep3B cells were trypsinized, seeded in 12-well plates at a concentration of 4 × 10^5^ cells/well, and grown overnight in a CO_2_ incubator. After 24 h, the media was removed from the wells, and a horizontal line was drawn across the bottom of the wells using an ultrafine tip black marker. One milliliter of phosphate-buffered saline (PBS) was added to each well, and a scratch was made vertically top to bottom using a 200-µL micropipette tip. Care was taken to keep the scratch as even as possible. The wells were washed twice with 1 mL PBS and the media changed to a reduced serum media (media with 5% fetal bovine serum). The plate was incubated in a CO_2_ incubator. Bright field images were taken using an ECHO revolve microscope at 4× magnification at 0, 4, 8, 24, and 48 h. The horizontal line in each well was used as a reference to obtain images from the same field across time points. A minimum of five images were taken per treatment group.

Images were analyzed in ImageJ (version 1.53k) using the wound healing size plugin developed specifically for the high-throughput analysis of wound healing assays [23]. The analysis settings (variance window radius, threshold value, and percentage of saturated pixels) were kept the same for all the images. The wound healing area obtained from ImageJ was exported and analyzed in Microsoft Excel 2016. The wound healing rate after treatment was calculated using the following formula:Wound healing rate (%) = (Wound area at 0 h − wound area at *n* hours/wound area at 0 h) × 100.

### 2.8. Quantification of Apoptosis by Flow Cytometry

HepG2 and Hep3B cells were treated with each of the Cjoc42 derivatives (AFM-1 and -2, JA-1–38 and DK-1–7) at a 25-μM concentration for 72 h. These cells, along with untreated control cells, were then harvested by the trypsin treatment and collected by spinning at 3000 rpm for 3 min. Apoptosis was measured using an Annexin V-APC)/propidium iodide (PI) Apoptosis detection kit (Biolegend; San Diego, CA, USA). Briefly, the cells were washed in a cell-staining buffer and resuspended in Annexin V-binding buffer. Following staining with 5 µL of Annexin V-APC and 8 µL propidium iodide, the cells were analyzed using Cytoflex flow cytometer (Beckman Coulter). In the dot plots generated, the lower left (LL) quadrant shows the percentage of healthy cells, the lower right (LR) quadrant shows the percentage of early apoptotic cells, and the upper left (UL)/upper right (UR) quadrants show the percentage of necrotic and late apoptotic cells [24,25]. This experiment was performed once with three readings, and the results were reconfirmed by performing an additional experiment to detect apoptosis using Apotracker Green (Biolegend).

### 2.9. Apoptosis Detection by Apotracker Green

HepG2 and Hep3B cells were treated with a 25-μM concentration of each of the three cjoc42 derivatives in a 96-well tissue culture plate. After 72 h of treatment, Apotracker green reagent (Biolegend) was added at a 0.2-μM concentration and imaged after 20 min under a microscope at 10× magnification. The experiment was done in triplicate and compared to the untreated controls. Image acquisition was done under the same laser power and gain for all samples. The total number of cells in each field were counted in the bright field mode, and the number of apoptotic cells in the same field were counted in the green fluorescence mode. The percent of apoptotic cells was then calculated.

### 2.10. Synergy Assays

Cells were treated in quadruplicate in a 96-well plate with increasing concentrations of cjoc42 derivatives (1–256 µM) and a chemotherapy agent (cisplatin (0.5–20 µM) or doxorubicin (0.01–1 µM for HepG2 and 0.01–0.75 µM for Hep3B)) as single agents or in combination. After 72 h of treatment, the MTT assay was performed. The data was run through Combenefit software to determine if synergy was present [25]. For the analysis, we utilized the Highest Single Agent (HSA) model. Synergy was depicted in blue and antagonism in red.

### 2.11. Statistical Analysis

All values were presented as the standard error of the mean. An unpaired Student’s *t*-test was applied for the comparison of normally distributed data. One-way ANOVA analysis was utilized with a Bonferroni test for multiple comparisons. Statistical significance was defined as: * = *p* < 0.05, ** = *p* < 0.01, and *** = *p* < 0.001. GraphPad PRISM (Version 8.4.3, California, USA) was used for all figures and statistical analyses. All experiments were performed in triplicate, unless specified in Section 2.

## 3. Results

### 3.1. Cjoc42 Derivatives Showed Antiproliferative Effects in Hep3B and HepG2 Cells

Our previous studies showed that cjoc42 inhibits the proliferation of HBL cells. Given the development of new cjoc42 derivatives, we examined the antiproliferative effects of three cjoc42 derivatives: AFM-1-2, DK-1–7, and JA-1–38 in the Hep3B and HepG2 cell lines using concentrations up to 25 μM (Figure 2A). We found that a statistically significant reduction in proliferation was observed at the 25-μM dose of all three compounds in both cell lines. Overall, there was a stronger antiproliferative effect seen in the Hep3B cells, with a statistically significant reduction at all concentrations. This was consistent with the IC50 dosing of these compounds (Figure 2B), with the exception of JA-1–38, which had a slightly lower IC50 concentration in HepG2 cells as compared to Hep3B (17.5 μM vs. 25.3 μM, respectively) [18]. We also measured the saturation effect, using up to 600-μM dosing (Figure 2C). In Hep3B, there was no significant change in the effect seen beyond 25 μM. In HepG2, there was a further decrease in the proliferation with AFM-1 and -2 and DK-1–7 at 125 μM but no changes with JA-1–38 beyond 25 μM.

### 3.2. Treatment with AFM-1 and -2 Resulted in the Rescue of Tumor Suppressor Proteins CUGBP1, HNF4α, and Rb and a Decrease in Cell Cycle Markers in HepG2

To evaluate the downstream effects of Gankyrin inhibition with AFM-1 and -2 on the key regulators of cell proliferation, we performed Western blots of the TSPs in both cell lines (Figure 3). In HepG2 cells, there was an increase in several TSPs, including CUGBP1, HNF4α, and Rb (Figure 3A,B). This increase was statistically significant for all three TSPs at the 15-μM dose and also for Rb at the 25-μM dose. In Hep3B cells, HNF4α and Rb both had a statistically significant increase at the 15-μM, 20-μM, and 25-μM concentrations (Figure 3C,D). There was no change in the protein levels of Gankyrin after treatment with AFM-1 and -2 (Appendix A). We also evaluated cell cycle markers cdc2 and ph-Rb. In HepG2 cells, there was a statistically significant decrease in cdc2 expression at the 25-μM concentration. Interestingly, we did see an increase in cdc2 at the 15-μM concentration, which was not expected given the results of the proliferation assay (Figure 2A). There was no difference in the Ph-Rb expression in either cell line or cdc2 expression in Hep3B cells seen in the protein analysis. 

Next, we examined the mRNA expression of TSPs in both the HepG2 and Hep3B cell lines (Figure 3E,F and Appendix A). Since Gankyrin works on the protein level with the degradation of TSPs by the ubiquitin proteasome system, we did not anticipate changes in the tumor suppressor mRNA levels, similar to what was previously shown with cjoc42 [8]. This hypothesis held true in Hep3B cells, with no difference in mRNA expression. However, in HepG2 cells, we observed a statistically significant decrease in the mRNA expression of CUGBP1 and HNF4α with AFM-1 and -2. Migration and cell invasiveness with AFM-1 and -2 was also examined in Hep3B cells (Appendix A). At 4, 8, and 48 h, there was a statistically significant reduction in wound healing for both the 20-μM and 25-μM doses. We did not detect a difference at 24 h, although there was a trend towards decreased wound healing at 20 μM.

### 3.3. Treatment of Cancer Cells with cjoc42 Derivatives DK-1–7 Demonstrated an Increase in Several Tumor Suppressor Proteins but Variable Changes in Cell Cycle Markers

Next, we performed a similar molecular analysis of the downstream effects after treatment with DK-1–7 (Figure 4). In HepG2 cells, we saw an increase in p53 and Rb expression. In Hep3B cells, we saw a dramatic increase in HNF4α at all concentrations. There was also an increase in Rb and CUGBP1. There was an increase in Gankyrin expression with the 15-μM treatment in the HepG2 cells but no significant changes in the Hep3B cells (Appendix A). Similar to AFM-1 and -2, there was a decrease in the mRNA levels of some tumor suppressors seen after treatment with DK-1–7 in the HepG2 cells (Figure 4E and Appendix A), but no statistically significant changes were seen in the Hep3B cells (Figure 4F and Appendix A). No changes in the proliferation markers (protein or mRNA) were seen in the HepG2 cells; however, we did see an increase in the protein expression of cdc2 and ph-Rb in Hep3B cells despite an antiproliferative effect by the proliferation assay (Figure 2A). Overall, our findings showed similar trends in tumor suppressor activity on the protein and mRNA levels, although we did not capture a decrease in the cell cycle markers despite a reduction in the proliferation by the MTT assay (Figure 1).

### 3.4. Treatment of Cancer Cells with JA-1–38 Demonstrated an Increase in Tumor Suppressor Proteins and a Decrease in Cell Cycle Markers

JA-1–38 was also analyzed using Western blotting and QRT-PCR (Figure 5). CUGBP1, Rb, and p53 were all increased in the treated HepG2 cells (Figure 5A,B). There was also an increase in HNF4α, but this did not reach statistical significance. There were no major differences in cdc2 or ph-Rb expression in these cells. In Hep3B cells, there was an increase in HNF4α expression for all three concentrations (Figure 5C,D). We did see an unexpected decrease in Rb expression at 20 μM but no changes at higher concentrations. There was also a decrease in cdc2 and ph-Rb expression at 20 and 25 μM, which was statistically significant for cdc2 at 25 μM. Gankyrin expression was unchanged in HepG2 cells and decreased in Hep3B cells treated at the 25-μM concentration (Appendix A). QRT-PCR demonstrated a decrease in CUGBP1 at 25 μM in HepG2 cells (Figure 5E) and no change in Hep3B cells (Figure 5F). There were no changes in the mRNA expression of other TSPs, Gankyrin, or cdc2 in either cell line (Figure 5E,F and Appendix A). Reviewing all the data for JA-1–38, there was a similar biological profile with an increase in TSPs, decrease in cell cycle markers (in Hep3B cells), and variable changes in mRNA expression between the two cell lines.

### 3.5. Treatment of Cancer Cells with AFM-1 and -2 Leads to a Reduction in Stem Cell Phenotype

Next, we evaluated the change in the stem cell markers and apoptosis after treatment with each cjoc42 derivative (Figure 6). To measure the stem cell phenotype, we analyzed the mRNA levels of CD133, CD13, KRT19, Oct4, and Thy1. We initially treated the cells with 25 μM of each compound and saw a decreased expression of the stem cell markers after treatment with AFM-1 and -2 and JA-1–38; however, most markers were not statistically significant (data not shown). To see if this was perhaps a dose-dependent phenomenon, we then treated cells with 64 μM and 128 μM of AFM-1 and -2 (Figure 6). In order to prevent excessive cell death, mRNA was isolated after only 24 h of treatment with the higher concentrations. These higher concentrations resulted in a statistically significant reduction in the stem cell phenotype in both cell lines. Specifically, there was a decrease in the expression of CD133, CD13, and KRT in both cell lines. In HepG2 cells, there was also a decrease in CD44 and Thy1, and, in Hep3B cells, a decrease in EPCAM.

### 3.6. An Increase in Apoptosis Was Detected in Hep3B Cells Treated with AFM-1 and -2

We next evaluated the cytotoxic effect of each of the cjoc42 derivatives by monitoring apoptosis (Figure 7). The cells were treated with 25 μM of each drug for 72 h prior to the analysis. Using Apotracker Green stain, the number of apoptotic cells was counted. In HepG2 cells, there was no significant change in apoptosis after treatment with any of the cjoc42 derivatives (Figure 7A,B). However, there was an increase in the apoptotic cells seen after AFM-1 and -2 treatment in Hep3B cells (Figure 7C,D). Flow cytometry for Annexin V confirmed these results, with no differences seen in HepG2 cells, and an increase with AFM-1 and -2 in Hep3B cells (Figure 7E–H). 

### 3.7. A Synergistic Effect Was Found between cjoc42 Derivatives and Doxorubicin, While an Antagonistic Effect Was Seen with Cisplatin

In order to understand if Gankyrin inhibition with these cjoc42 derivatives improved the chemosensitivity, we performed synergy assays with cisplatin and doxorubicin. In HepG2 cells, synergy was seen between all three cjoc42 derivatives and doxorubicin (Figure 8A). Synergy was seen with most doses of AFM-1 and -2 ranging from 64 to 256 μM and doxorubicin doses of 0.25 to 1 μM. With cisplatin, however, there was no synergy seen, and instead, some mild antagonism was noted, especially with higher doses of cisplatin (Figure 8B). We found a similar result in the Hep3B cells. Specifically, drug synergy was seen between all three cjoc42 derivatives and doxorubicin (Figure 8C), most notably when the drugs were treated with 64 μM of a cjoc42 derivative. Within that dosing schema, the synergy was strongest with 0.25 μM of doxorubicin compared to higher doses of doxorubicin. In combination with the data from the HepG2 cells, this suggests that, in the presence of Gankyrin inhibition, a lower dose of doxorubicin may be sufficient and perhaps more favorable in achieving synergy and maximizing the cytotoxicity. Synergy was not reliably seen with cisplatin, and more importantly, a mild antagonism was seen with higher doses of cisplatin (15–20 μM) in combination with a cjoc42 derivative (Figure 8D).

The synergy between Gankyrin inhibition and doxorubicin is associated with the nuclear transport of doxorubicin by the 20S proteasome. Doxorubicin leads to cell death through DNA damage, which requires nuclear localization of the drug. Previous studies have demonstrated that doxorubicin enters the nucleus by binding to the 20S subunit of the 26S proteasome [26]. To understand if this is the mechanism behind the synergy with cjoc42 derivatives, we performed Western blots for the α subunits (α1–7) of the 20S proteasome within the cytoplasmic and nuclear extracts after treatment (Figure 9). A dosing strategy was based on the results of the previously described synergy assays. For HepG2 cells, this was 64 μM of AFM-1 and -2 with 0.25 μM of doxorubicin and, for Hep3B cells, 64 μM AFM-1 and -2 and 0.5 μM of doxorubicin. The protein was isolated after 48 h of treatment in the HepG2 cells and 24 h in the Hep3B cells (due to the high cell death at the 48-h mark). β-actin and HDAC were used as loading controls for the cytoplasmic and nuclear extracts, respectively. In HepG2 cells, there was a statistically significant decrease in the expression of α proteasome in the cytoplasm and a trend towards an increase in the nuclear expression (Figure 9A,B). There were no changes seen in the cytoplasmic expression of Hep3B cells, but an increase in nuclear expression with a combination treatment was observed (Figure 9C,D).

## 4. Discussion

The use of targeted agents in cancer therapy has become a more common practice, with the benefit of directly killing malignant cells and minimizing off-target effects. In order to develop these compounds, there must be an understanding of what pathways drive cancer proliferation and, as such, would be targetable. In pediatric liver cancer, the use of targeted therapy is limited to sorafenib, a multi-kinase inhibitor that delays progression by 3 months in HCC [27]. With an overall survival of 70% in HBL and 20% in HCC, it is clear newer therapy options need to be pursued, and targeted agents should be investigated [2].

Previous studies have found Gankyrin as a driver oncoprotein in the proliferation of pediatric liver cancer [12,28], making it an attractive target. Cjoc42 was the first compound developed to inhibit Gankyrin, and we previously demonstrated that this compound works by binding directly to Gankyrin, leading to a change in Gankyrin’s structure and impairing its ability to bind to and degrade TSPs [8]. However, high IC50 values made this compound unattractive for clinical use [8]. More recently, cjoc42 derivatives have been developed with a better affinity for Gankyrin and improved IC50 values (Figure 2). The purpose of our study was to analyze the in vitro effects of treatments with three cjoc42 derivatives: AFM-1 and -2, DK-1–7, and JA-1–38 in the HepG2 (HBL) and Hep3B (HCC) cell lines. We completed a biological assessment to evaluate the changes in tumor suppressors, cell cycle markers, stem cell markers, and apoptosis (Figure 1). Given that targeted agents typically require a combination with chemotherapy to maximize their cytotoxic effect, we also performed synergy assays with cisplatin and doxorubicin, which are commonly used in both HBL and HCC therapy.

Overall, there was a statistically significant increase in at least two TSPs after treatment with any of the three agents in either cell line. This suggests that cjoc42 derivatives inhibit the protein–protein interactions between Gankyrin and certain TSPs that would otherwise allow for the degradation of these proteins and thereby shift the balance towards cancer proliferation. While an antiproliferative effect was demonstrated (Figure 2), changes in cell cycle markers cdc2 and ph-Rb were not consistently detected. Specifically, there was only a significant decrease in at least one cell cycle marker with AFM-1 and -2 in HepG2 and JA-1–38 in Hep3B.

At an mRNA level, we did not see any changes in the levels of the tumor suppressors with any compound in the Hep3B cells. Since Gankyrin works on the protein level, this was an expected result and consistent with what was previously shown with cjoc42 [8]. Interestingly, there was a >2-fold decrease in the mRNA levels of CUGBP1 (Figure 3E, Figure 4E, and Figure 5E), as well as a few other tumor suppressor genes (Appendix A), in HepG2 cells. The majority of these changes in mRNA were also associated with a significant increase in protein expression, suggesting this may be a negative feedback loop.

We also wanted to analyze other biological alterations that may be seen after treatment with cjoc42 derivatives, including changes in the stem cell phenotype and apoptosis (Figure 6 and Figure 7, respectively). We examined stem cell markers CD133, CD13, KRT, EPCAM, CD44, Thy1, and Oct4. Initially, these markers were analyzed after treatment with 25 μM of each compound. While there was a decrease in several markers, in most instances, it was a nonsignificant trend. In order to see if this was dose-dependent, we then analyzed them after treatments with 64 and 128 μM of each compound (Figure 6). The results supported a dose-dependent alteration in the stem cell phenotype, with a significant reduction in the majority of the stem cell markers seen at the 64-μM or 128-μM dose. The apoptosis was evaluated by Apotracker green staining and flow cytometry for Annexin V (Figure 7). There was a significant increase in the apoptosis in Hep3B cells after treatment with AFM-1 and -2; however, there were no changes seen in HepG2 or with other treatment in Hep3B. These results suggest the presence of alternate cell death pathways triggered by these drugs.

Synergy with doxorubicin and cisplatin was performed with each of the three cjoc42 derivatives (Figure 8). All compounds demonstrated synergy with doxorubicin in both cell lines. Doxorubicin exerts its effects by intercalations in DNA with the subsequent inhibition of DNA synthesis and generation of free radicals [29]. In order to enter the nucleus, doxorubicin will bind to the 20S subunit of the 26S proteasome [30]. We hypothesized that because cjoc42 derivatives bind to Gankyrin, which is part of the 19S regulatory subunit of the 26S proteasome, the 20S subunit becomes more available to transport doxorubicin to the nucleus. To test this hypothesis, we measured the nuclear and cytoplasmic protein expressions of the α1–7 subunits of the 20S proteasome after treatment with AFM-1 and -2, doxorubicin, or the drug combination (Figure 9). In HepG2 cells, we saw a statistically significant decrease in the cytoplasmic expression and trend towards increased nuclear expression of the α subunit with the combination treatment. In Hep3B cells, there were no changes in the cytoplasmic expression, but there was an increase in the nuclear expression with the combination treatment. This data suggests that, in the presence of cjoc42 derivatives, the 20S proteasome was more accessible to transport doxorubicin into the nucleus and exert its cytotoxic effects. With cisplatin, there were no changes seen overall, although mild antagonism was observed in higher doses. Since cisplatin is known to affect the methylation status in other types of cancer [31], we assessed the changes in global methylation between different treatment groups and saw no changes (data not shown). Further mechanistic studies may be considered if significant antagonism is seen with other Gankyrin inhibitors.

Gankyrin overexpression is a common theme seen in many solid tumors, making it an attractive therapeutic approach. In pediatric liver cancer, Gankyrin inhibition could be considered as part of upfront HCC and relapsed HBL therapy, both of which have diminished prognoses. We did see a variability in the results between cell lines, which speaks to the inherent biological differences between HBL and HCC. As the use of individualized, precision-based medicine becomes more commonplace, evaluating tumors for Gankyrin expression by immunohistochemistry and/or molecular analysis may help identify patients most likely to benefit from this therapeutic strategy. Additionally, combination therapy is often more effective than single agents, and our results do support synergy between doxorubicin and second-generation cjoc42 derivatives.

## 5. Conclusions

Second-generation inhibitors of Gankyrin, also known as cjoc42 derivatives, are a promising therapeutic strategy for children with liver cancer. These drugs work by inhibiting the function of Gankyrin, leading to an increase in the tumor suppressor proteins that shift the balance away from uncontrolled proliferation. The results were variable between the two cell lines, which reflects the inherent biological differences between hepatoblastoma and hepatocellular carcinoma. Overall, AFM-1 and -2 showed the most promising effects, with a rescue of the TSPs, reduction in the cell cycle markers (in HepG2 cells), decrease in stem cell expression (in Hep3B cells), and increase in apoptosis (in Hep3B cells). All three compounds demonstrated synergy with doxorubicin by aiding in the nuclear localization of this chemotherapeutic agent and mild antagonism with cisplatin. These findings support Gankyrin inhibition as a therapeutic model and highlight the importance of preclinical testing due to the varied effects seen with combination therapy.

## Figures and Tables

**Figure 1 cancers-14-03068-f001:**
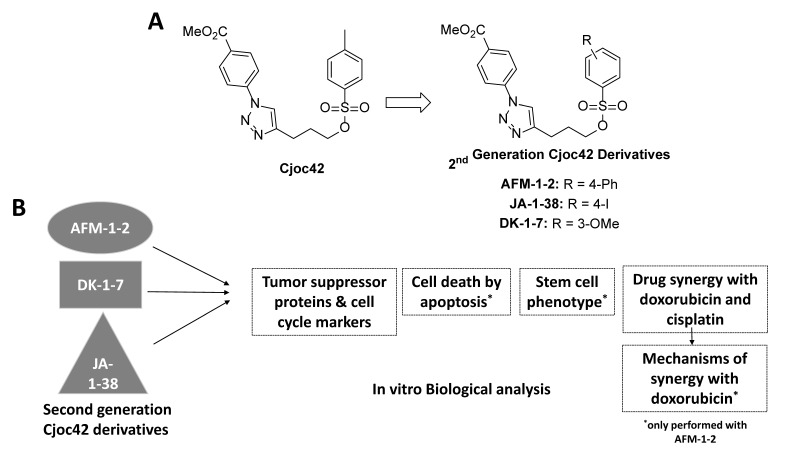
Review of Cjoc42 derivatives and the experimental design. (**A**) Chemical structure of the cjoc42 derivatives. (**B**) Graphical illustration of the experimental design with all three cjoc42 derivatives.

**Figure 2 cancers-14-03068-f002:**
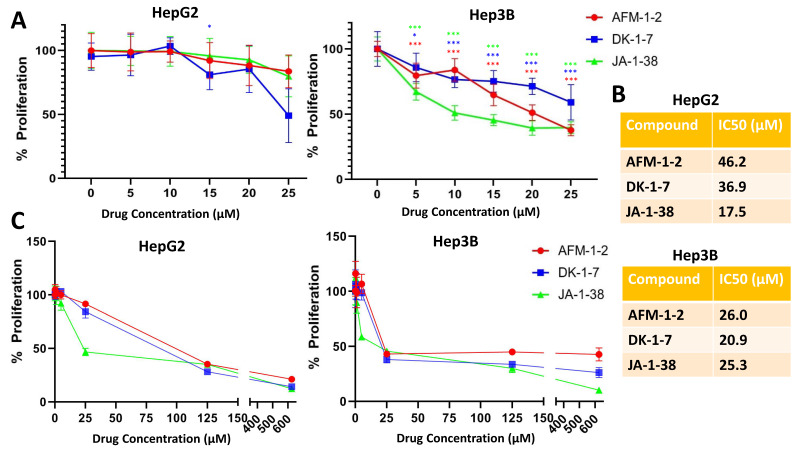
Cjoc42 derivatives and their antiproliferative effects. (**A**) Proliferation assay of three cjoc42 derivatives at 5–25 μM doses after 72 h of treatment in HepG2 cells and Hep3B cells. In HepG2 cells (**left**), a statistically significant difference was found in DK-1–7 at the 15-μM dose (*p* = 0.031) and for all three compounds at the 25-μM dose (AFM-1 and -2, *p* = 0.025; DK-1–7, *p* < 0.001; JA-1–38, *p* = 0.022). In Hep3B cells (**right**), a statistically significant difference in proliferation was seen at all concentrations of AFM-1 and -2 (5 μM, <0.001; 10 μM, <0.001; 15 μM, <0.001; 20 μM, <0.001; 25 μM, <0.001); DK-1–7 (5 μM, =0.035; 10 μM, <0.001; 15 μM, <0.001; 20 μM, <0.001; 25 μM, <0.001); and JA-1–38 (5 μM, <0.001; 10 μM, <0.001; 15 μM, <0.001; 20 μM, <0.001; 25 μM, <0.001). * = *p* < 0.05, and *** = *p* < 0.001 (**B**) IC50 values for each compound in the HepG2 and Hep3B cells. (**C**) Proliferation assay with higher concentrations of each compound in the HepG2 and Hep3B cells (up to 600 μM).

**Figure 3 cancers-14-03068-f003:**
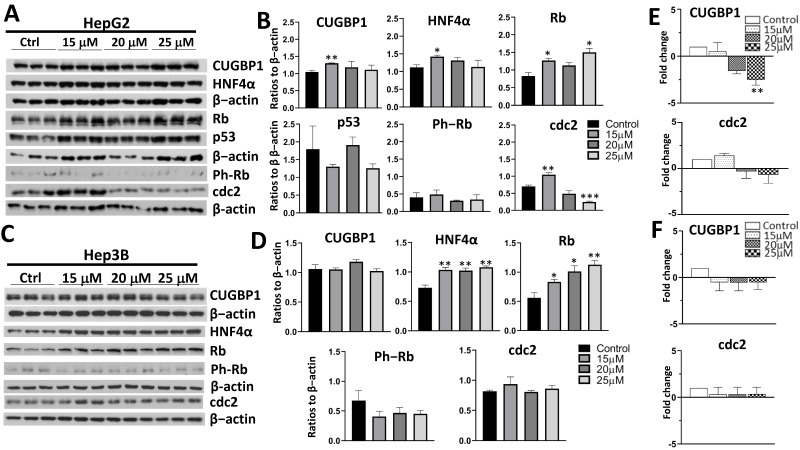
Treatment with AFM-1 and -2 resulted in the rescue of tumor suppressor proteins CUGBP1, HNF4α and Rb, a decrease of the cell cycle markers in HepG2. (**A**) Western blots of HepG2 cells treated with increasing concentrations of AFM-1 and -2. (**B**) Quantitation of the Western blot results of HepG2 cells treated with increasing concentrations of AFM-1 and -2. There was a statistically significant increase in CUGBP1 expression at 15 μM (*p* = 0.009), HNF4α at 15 μM (*p* = 0.023), and Rb at 15 μM and 25 μM (*p* = 0.018 and *p* = 0.011, respectively). There was an increase in cdc2 expression at 15 μM (*p* = 0.007) but a decrease at 20 μM and 25 μM (*p* = 0.091 and *p* < 0.001, respectively). * = *p* < 0.05, ** = *p* < 0.01, and *** = *p* < 0.001. (**C**) Western blots of Hep3B cells treated with increasing concentrations of AFM-1 and -2. (**D**) Quantitation of the Western blot results of Hep3B cells treated with increasing concentrations of AFM-1 and -2. The TSPs HNF4α and Rb showed a statistically significant increase in expression at the 15-μM, 20-μM, and 25-μM concentrations (HNF4α: 15 μM, *p* = 0.009; 20 μM, *p* = 0.01; 25 μM, *p* = 0.003; Rb: 15 μM, *p* = 0.047; 20 μM, *p* = 0.026; 25 μM, *p* = 0.007). * = *p* < 0.05, and ** = *p* < 0.01. (**E**) QRT-PCR of HepG2 cells treated with AFM-1 and -2. A statistically significant decrease in CUGBP1 mRNA expression was seen at 25 μM (*p* = 0.003). There was no difference in cdc2. ** = *p* < 0.01. (**F**) QRT-PCR of Hep3B cells treated with AFM-1 and -2. No difference was seen in mRNA expression of CUGBP1 or cdc2.

**Figure 4 cancers-14-03068-f004:**
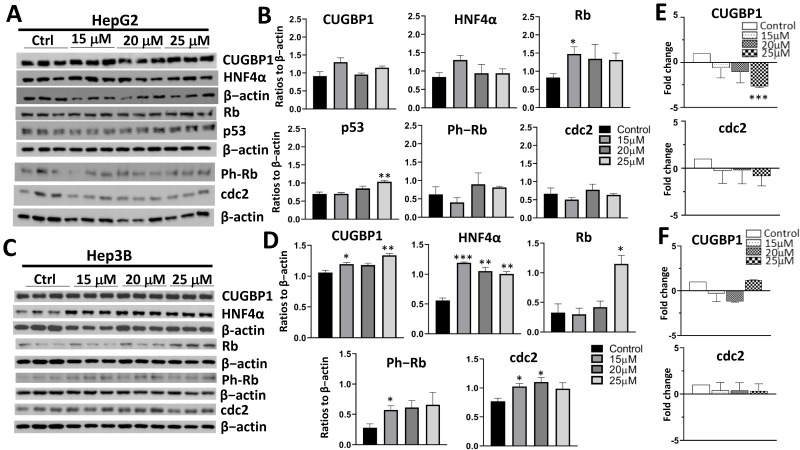
Treatment of cancer cells with cjoc42 derivative DK-1–7 demonstrated an increase in several tumor suppressor proteins but variable changes in cell cycle markers. (**A**) Western blots of HepG2 cells treated with increasing concentrations of DK-1–7. (**B**) Quantitation of the Western blot results of HepG2 cells treated with increasing concentrations of DK-1–7. There was an increase in p53 expression at 25 μM (*p* = 0.006) and Rb at 15 μM (*p* = 0.045). * = *p* < 0.05, ** = *p* < 0.01. (**C**) Western blots of Hep3B cells treated with increasing concentrations of DK-1–7. (**D**) Quantitation of the Western blot results of Hep3B cells treated with increasing concentrations of DK-1–7. There was a profound increase in HNF4α at all concentrations (15 μM, *p* < 0.001; 20 μM, *p* = 0.003; 25 μM, *p* = 0.002) and Rb at 25 μM (*p* = 0.017). CUGBP1 was also increased with treatment at 15 μM (*p* = 0.038) and 25 μM (*p* = 0.004). * = *p* < 0.05, ** = *p* < 0.01, and *** = *p* < 0.001. (**E**) QRT-PCR of HepG2 cells treated with DK-1–7. There was a statistically significant decrease in CUGBP1 mRNA expression at 25 μM (*p* < 0.001) but no change in cdc2. *** = *p* < 0.001. (**F**) QRT-PCR of Hep3B cells treated with DK-1–7. No changes seen in CUGBP1or cdc2 mRNA expression.

**Figure 5 cancers-14-03068-f005:**
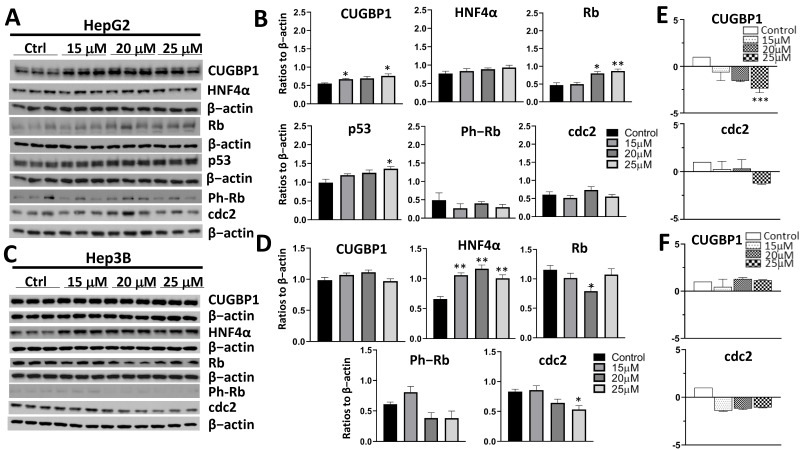
Treatment of cancer cells with JA-1–38 demonstrated an increase in tumor suppressor proteins and a decrease in cell cycle markers. (**A**) Western blots of HepG2 cells treated with increasing concentrations of JA-1–38. (**B**) Quantitation of the Western blot results of HepG2 cells treated with increasing concentrations of JA-1–38. CUGBP1 was elevated at 15 μM (*p* = 0.015) and 25 μM (*p* = 0.018), Rb at 20 μM (*p* = 0.015) and 25 μM (*p* = 0.01), and p53 at 25 μM (*p* = 0.029). HNF4α showed a trend of increased expression with higher doses of JA-1–38, but this did not reach statistical significance. * = *p* < 0.05, ** = *p* < 0.01. (**C**) Western blots of Hep3B cells treated with increasing concentrations of JA-1–38. (**D**) Quantitation of the Western blot results of Hep3B cells treated with increasing concentrations of JA-1–38. There was an increase in HNF4α at all three doses (15 μM, *p* = 0.002; 20 μM, *p* = 0.002; 25 μM, *p* = 0.008). Rb was significantly decreased at 20 μM (*p* = 0.026) but otherwise unchanged at the other two doses. Cdc2 expression was decreased at 25 μM (*p* = 0.016). pRb showed a trend of reduced expression at 20 and 25-μM dosing but was not statistically significant. * = *p* < 0.05, ** = *p* < 0.01. (**E**) QRT-PCR of HepG2 cells treated with JA-1–38. There was a significant decrease in CUGBP1 expression at 25 μM (*p* = 0.001) but no changes in cdc2. *** = *p* < 0.001. (**F**) QRT-PCR of Hep3B cells treated with JA-1–38. No difference was seen in the mRNA expression of CUGBP1 or cdc2.

**Figure 6 cancers-14-03068-f006:**
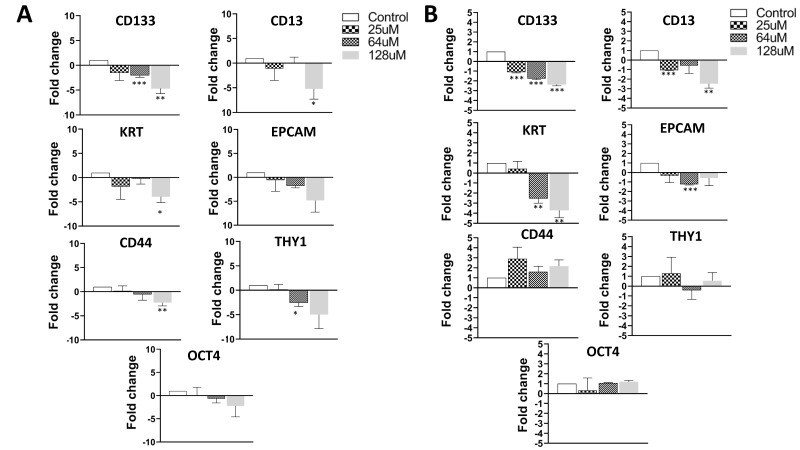
Treatment of cancer cells with AFM-1 and -2 leads to a reduction in the stem cell phenotype. (**A**) QRT-PCR results of the expression of stem cell markers in HepG2 cells treated with 25 μM, 64 μM, and 128 μM. A significant reduction in the expression of CD133 was seen with 64 and 128-μM dosing (*p* = 0.001 and 0.006, respectively). There was also a decrease in CD13 and KRT seen at 128 μM (*p* = 0.040 and 0.014, respectively) and Thy1 at 64 μM (*p* = 0.011). * = *p* < 0.05, ** = *p* < 0.01, and *** = *p* < 0.001. (**B**) QRT-PCR results of the expression of stem cell markers in Hep3B cells treated with 25 μM, 64 μM, and 128 μM. A significant reduction in the expression of CD133 was seen with all three doses (25 μM, *p* = ≤0.001; 64 μM, *p* = <0.001; 128 μM, *p* = <0.001). The CD13 expression was reduced at 25 μM (*p* = <0.001) and 128 μM (*p* = <0.002). The KRT expression was reduced at 64 and 128 μM (*p* = 0.002 and 0.003, respectively) and EPCAM at 64 μM (*p* = <0.001). ** = *p* < 0.01 and *** = *p* < 0.001.

**Figure 7 cancers-14-03068-f007:**
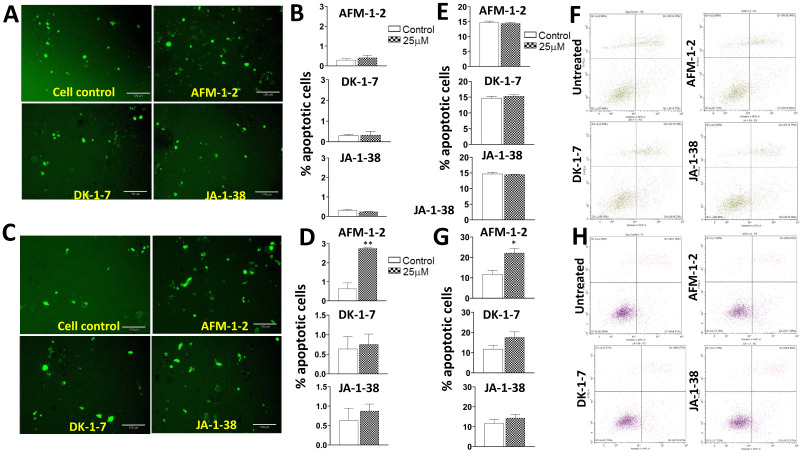
An increase in apoptosis was detected in Hep3B cells treated with AFM-1 and -2. (**A**) Microscope images of Apotracker Green-stained HepG2 cells treated with AFM-1 and -2, DK-1–7, and JA-1–38. (**B**) Percentage of apoptotic cells stained by Apotracker Green in HepG2 treated with AFM-1 and -2, DK-1–7, and JA-1–38. (**C**) Microscope images of Apotracker Green-stained Hep3B cells treated with AFM-1 and -2, DK-1–7, and JA-1–38. (**D**) Percentage of apoptotic cells stained by Apotracker Green in Hep3B cells treated with AFM-1 and -2, DK-1–7, and JA-1–38. The treatment with AFM-1 and -2 resulted in a significant increase in apoptosis compared to the control cells (*p* = 0.002). ** = *p* < 0.01. (**E**) Percentage of apoptotic cells evaluated by Annexin V-APC/PI staining in HepG2 cells. Data are presented as the mean percentage of measured apoptotic cells. (**F**) Representative dot plots of HepG2. (**G**) Percentage of apoptotic cells evaluated by Annexin V-APC/PI in Hep3B cells. Data are presented as the mean percentage of measured apoptotic cells. The treatment with AFM-1 and -2 resulted in a significant increase in apoptosis compared to the control cells (*p* = 0.022). * = *p* < 0.05. (**H**) Representative dot plots of Hep3B.

**Figure 8 cancers-14-03068-f008:**
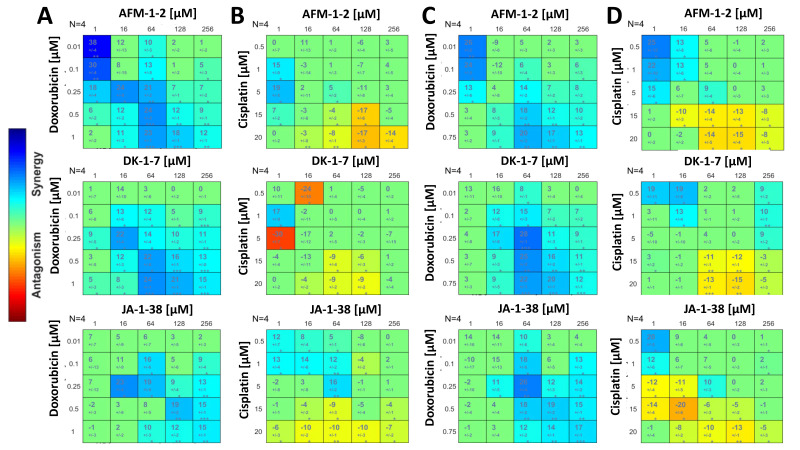
A synergistic effect was found between cjoc42 derivatives and doxorubicin, while an antagonistic effect was seen with cisplatin. (**A**) Dose–response plots of doxorubicin with AFM-1 and -2, DK-1–7, and JA-1–38 in the HepG2 cell line. (**B**) Dose–response plots of cisplatin with AFM-1 and -2, DK-1–7, and JA-1–38 in the HepG2 cell line. (**C**) Dose–response plots of doxorubicin with AFM-1 and -2, DK-1–7, and JA-1–38 in the Hep3B cell line. (**D**) Dose–response plots of cisplatin with AFM-1 and -2, DK-1–7, and JA-1–38 in the Hep3B cell line. Synergy is depicted in blue and antagonism in red.

**Figure 9 cancers-14-03068-f009:**
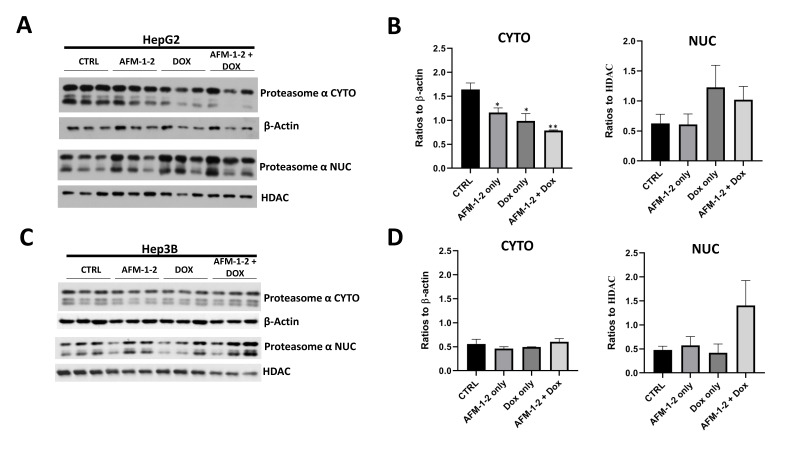
Synergy between Gankyrin inhibition and doxorubicin is related to the nuclear transport of doxorubicin by the 20S proteasome. (**A**) Western blot results of proteasome α in the nuclear and cytoplasmic extracts of HepG2 cells treated with AFM-1 and -2 alone, doxorubicin (abbreviated as Dox in the figure) alone, or in combination. (**B**) Quantitation of proteasome α in the nuclear and cytoplasmic extracts of HepG2 cells treated with AFM-1 and -2 alone, doxorubicin alone, or in combination. A statistically significant decrease in the cytoplasmic expression was seen with AFM-1 and -2 only (*p* = 0.044), Dox only (*p* = 0.032), and AFM-1 and -2 + Dox (*p* = 0.003). There was an increase in the nuclear expression with Dox only and AFM-1 and -2 + Dox; however, it did not reach statistical significance (*p* = 0.210 and *p* = 0.220, respectively). * = *p* < 0.05 and ** = *p* < 0.01. (**C**) Western blot results of proteasome α in the nuclear and cytoplasmic extracts of Hep3B cells treated with AFM-1 and -2 alone, doxorubicin alone, or in combination. (**D**) Quantitation of proteasome α in the nuclear and cytoplasmic extracts of Hep3B cells treated with AFM-1 and -2 alone, doxorubicin alone, or in combination. There were no differences in the cytoplasmic expression in Hep3B cells and a trend of increasing in the nuclear expression with AFM-1 and -2 + Dox (*p* = 0.153).

## Data Availability

The data presented in this study are available within the article and the Appendix A.

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
