# Peer review of "Second Generation Small Molecule Inhibitors of Gankyrin for the Treatment of Pediatric Liver Cancer"

_cancers, 2022, doi:10.3390/cancers14133068_

Round 1

Reviewer 1 Report

D’Souza et al, in the manuscript Second Generation Small Molecule Inhibitors of Gankryin for the Treatment of Pediatric Liver Cancer”, investigate the effects in liver cancer cell lines of three inhibitor compounds (cjoc42 derivatives) of Gankyrin and the drug cooperation between cjoc42 derivatives and the chemotherapeutics commonly used in liver cancer.

As described by scientific literature, Gankyrin is a driver overexpressed oncoprotein that promote liver cancer by destroying tumor suppressor proteins (TSPs) in pediatric liver cancer. Cjoc42 was the first compound developed to inhibit Gankyrin. The second generation of Gankyrin inhibitors is based on modifications to the cjoc42 structure and resulted in a better affinity for Gankyrin and in an improved cytotoxicity.

The authors analyse the effects of treatment with three cjoc42 derivatives (AFM-1-2, DK-1-7, and JA-1-38) in HepG2 (hepatoblastoma, HBL) and Hep3B (hepatocellular carcinoma, HCC) cell lines and evaluate changes in tumor suppressors, cell cycle markers, stem cell markers, and apoptosis. Furthermore, in order to maximize the cytotoxic effect of these compounds, they also performed synergy assays with cisplatin and doxorubicin, which are commonly used in both HBL and HCC therapy.

They demonstrated:

  • A significantly reduced cell proliferation in liver cancer cells treated with these compounds by preventing the degradation of tumor suppressor proteins.
  • The enhancing effects of cjoc42 derivatives in combination to

Finally, the authors suggest the promising role of small molecule inhibitors of Gankryin as a therapeutic model, proposing combinatorial treatments of them with doxorubicin.

The article is well-developed, and the contents are described in detail. In support, the bibliography is substantial and adequate.

However, this referee has the following observations:

-            Please, check in the Supplemental Figure 1 the unrevealed upregulation of Gankyrin after treatment with AFM-1-2 at 15µ. Is it correct its no significant expression?

 -            Verify and use the same order of western blots and graphs in all figures (i.e. in western blot in fig 2 A e B the order of the proteins is different respect to fig. 3 and further in figure 4).

 -            Synergistic effect reported in figure 7 is poor understandable. Characters are not clearly visible. Please modify.

 -            In the discussion section, the authors explain that the synergy effect with Doxorubicin depends by its binding to the 20S subunit of the 26S proteosoma. On the other hand, no explanation has reported for the mild synergic effect revealed in the presence of cisplatin. Please introduce a comment at this reguard.

Reviewer 2 Report

The authors have tested three different second-generation small molecule inhibitors of Gankryin for treating pediatric liver cancer. They also tested two different cell lines in this study.

Cancer relapse and resistance to the treatments is a big challenge clinically. This study has discussed the effect of these three cjoc42 derivatives in detail. However, there are a few concerns that need to be addressed.

  1. Cells behave differently in 2D vs. 3D. It’d be more convincing to test one of the G-inhibitors in a 3D organoids model. Also, I’d encourage authors to discuss the results from the clinical perspective as well.
  2. I am curious to see the IC50 result for this study for all the derivatives. Please make a graphical representation of the same.
  3. I suggest the authors perform a wound-healing assay to test the invasiveness/migration of HepG2 and Hep3B before and after treatment.
  4. Can this method help to develop cancer precision treatment?
  5. Why was there a significant variation in the treatment between the cell lines? Please discuss.
  6. Was there any saturation effect observed in the two cells under these three derivatives? Please discuss.
  7. Line 217 “…but no changes were seen in the Hep3B cells” – but the figure shows variations in the CUGBP1 mRNA levels. Please change the language to reflect the graphs.
  8. Does the inhibitor concentration affect healthy cells?
  9. All figures need attention. The legends and axis labels were not legible in any figure. Also, please expand the figure description to explain each section in the figure.
  10. I’d suggest the authors split figure 1 into two figures—figure 1 A and C as one figure; Figure 1B into another.
  11. Minor comment: Line 194: Authors mentioned the cdc2 at 15 uM concentration was not an expected result. Why? Please explain. This pattern was found in a few places – one more example – line 219.

Round 2

Reviewer 2 Report

The authors have addressed all the concerns.